# Accurate Reference Gas Mixtures Containing Tritiated Molecules: Their Production and Raman-Based Analysis

**DOI:** 10.3390/s21186170

**Published:** 2021-09-14

**Authors:** Simon Niemes, Helmut H. Telle, Beate Bornschein, Lucian Fasselt, Robin Größle, Florian Priester, Magnus Schlösser, Michael Sturm, Stefan Welte, Genrich Zeller

**Affiliations:** 1Tritium Laboratory Karlsruhe (TLK), Institute for Astroparticle Physics (IAP), Karlsruhe Institute of Technology (KIT), Hermann-von-Helmholtz-Platz 1, 76021 Karlsruhe, Germany; beate.bornschein@kit.edu (B.B.); lucian.fasselt@freenet.de (L.F.); robin.groessle@kit.edu (R.G.); florian.priester@kit.edu (F.P.); magnus.schloesser@kit.edu (M.S.); Michael.sturm@kit.edu (M.S.); stefan.welte@kit.edu (S.W.); genrich.zeller@kit.edu (G.Z.); 2Departamento de Química Física Aplicada, Campus de Cantoblanco, Universidad Autónoma de Madrid, 28049 Madrid, Spain; 3Institut für Physik, Humboldt-Universität zu Berlin, Unter den Linden 6, 10099 Berlin, Germany

**Keywords:** Raman spectroscopy, tritiated molecules, gas mixing, β-induced self-equilibration, tritiated reference samples

## Abstract

Highly accurate, quantitative analyses of mixtures of hydrogen isotopologues—both the stable species, H_2_, D_2_, and HD, and the radioactive species, T_2_, HT, and DT—are of great importance in fields as diverse as deuterium–tritium fusion, neutrino mass measurements using tritium β-decay, or for photonuclear experiments in which hydrogen–deuterium targets are used. In this publication we describe a production, handling, and analysis facility capable of fabricating well-defined gas samples, which may contain any of the stable and radioactive hydrogen isotopologues, with sub-percent accuracy for the relative species concentrations. The production is based on precise manometric gas mixing of H_2_, D_2_, and T_2_. The heteronuclear isotopologues HD, HT, and DT are generated via controlled, in-line catalytic reaction or by β-induced self-equilibration, respectively. The analysis was carried out using an in-line intensity- and wavelength-calibrated Raman spectroscopy system. This allows for continuous monitoring of the composition of the circulating gas during the self-equilibration or catalytic evolution phases. During all procedures, effects, such as exchange reactions with wall materials, were considered with care. Together with measurement statistics, these and other systematic effects were included in the determination of composition uncertainties of the generated reference gas samples. Measurement and calibration accuracy at the level of 1% was achieved.

## 1. Introduction

Quantitative, compositional analysis of gas mixtures, which contain hydrogen (H_2_), is frequently required in the characterization and monitoring of, for example, combustion gases. Much less common are mixtures that contain any of the other hydrogen isotopologues (D_2_, T_2_, HD, HT, and DT, in addition to H_2_). However, there are a number of research fields and practical applications requiring highly accurate analysis of such mixtures, specifically those which contain tritium-substituted species. This need for tritium analysis and monitoring is encountered in the following three topical applications.

Firstly, analysis of tritium purity and of operational mixture preparation is key in tritium production and processing installations, such as, for example, our own Tritium Laboratory Karlsruhe (TLK) [1].

Secondly, tritium/deuterium mixtures will be used as the fuel in future fusion reactors, such as ITER (the International Thermonuclear Experimental Reactor) and DEMO (the future Demonstration Power Plant), see Ref. [2]. The accurate analysis of the isotopic composition within the closed fuel cycle is of utmost importance for obvious operational and safety reasons.

Thirdly, high-purity T_2_ is used in the gaseous tritium β-source in the Karlsruhe Tritium Neutrino Experiment (KATRIN) [3]. The other isotopologues are present as low-level impurities but, nevertheless, have to be monitored accurately in order to minimize systematic effects on the measurement of the electron-neutrino mass (aiming at 0.2 eV/c^2^ sensitivity) [4].

Finally, it is noteworthy that tritium analysis and monitoring also is relevant in various biomedical (e.g., monitoring of tritium in bio-fluids), environmental (e.g., monitoring of tritium in river water and sediments), and industrial (e.g., detritiation of coolants in heavy water reactors) applications. However, by and large, the materials and samples encountered there are mostly in liquid or solid form, and tritium concentrations are normally at trace levels; both are therefore outside the scope of this publication.

While quite a few analytical methods have been, and are, used routinely in these applications—including ionization chambers, β-induced X-ray spectroscopy, gas chromatography, and mass spectrometry, amongst others (see Ref. [5])—it is Raman spectroscopy that has emerged as a method of choice for in-line gas analysis, being a non-contact, multi-species analysis technique with high sensitivity [6]. For example, KATRIN successfully employs Raman spectroscopy monitoring for continuous, in-line compositional analysis of its circulating source gas [7]. For the accountancy of tritium in the fusion fuel cycle of ITER/DEMO, Raman spectroscopy has been evaluated as one possible analytical tool [8,9,10], which is still partially under procurement [11], or is under development but has not been tested with tritium yet [12].

Regardless of the state of development and actual use, any Raman analysis system needs to be carefully calibrated to yield reliable compositional measurement results, with adequate accuracy (note that throughout this publication we will use the terminology of “precision”, “trueness”, and “accuracy” as defined in Ref. [13]).

As will be discussed in more detail in Section 2, in order to verify and calibrate the measurement results it is paramount to have standard samples available and/or measurement equipment whose method and performance can be linked to accepted standards or certification procedures. In general, accurate homogeneous calibration gas mixtures can be produced according to ISO standards 6142/6145 [14,15]. However, the two methods described therein cannot be used for species that are “potentially interactive substances” and “can decompose” [14]; hydrogen isotopologues belong to these categories. Already, samples containing only H_2_ and D_2_ will undergo (mostly surface-mediated) exchange reactions, and ultimately the sample will find its chemical equilibrium between H_2_, D_2_, and HD; but these exchange reactions are normally extremely slow at ambient temperatures (of the order of many weeks). In this context, it should be noted that exchange reactions in tritiated mixtures, which are linked to β-induced reaction-chain processes, lead toward chemical equilibrium on the time scale of only hours to a few days; during these periods, non-equilibrium between constituents is encountered, changing the gas composition significantly early on in any mixing process.

A few years ago, we developed a hydrogen–deuterium mixing (HYDE) loop to generate accurate reference mixture samples of H_2_/HD/D_2_ in chemical equilibrium, for a range of specified ratios [16]. Using these, we cross-validated the initial quantitative calibration of our laser Raman system used in the KATRIN gas analysis [17]. However, it should be noted that HYDE could only provide mixture calibration values for the stable isotopologues, but not the radioactive species, T_2_, DT, and HT. Based on the good agreement between measurements using well-defined HYDE gas mixtures for H_2_, HD, and D_2_, and experimentally confirmed theoretical Raman transition strengths [18], it was only conjured that the values for T_2_, DT, and HT would be equally reliable.

Thus, the concept for a tritium–hydrogen–deuterium (TRIHYDE) facility was developed, with the aim of generating gas mixtures containing defined concentrations of H_2_, HD, D_2_, T_2_, DT, and HT. The original intent was to independently confirm the crucial calibration factors for KATRIN, using said accurately mixed gas samples, now including the radioactive isotopologues. The setup and use of this system are described in this publication.

The primary goals of the studies described here were threefold, namely (i) to enable gas sample production of any combination of all six hydrogen isotopologues in chemical equilibrium and, as an application of these, (ii) to verify/cross-calibrate the theoretical Raman intensities versus accurate gas mixtures of tritiated hydrogen isotopologues. In addition, (iii) the overall measurement and calibration uncertainties were to be reduced, with respect to the predecessor experiment HYDE. Throughout this work, we have attempted to follow, as closely as possible, accepted calibration procedures and link our measurements to established methodologies and standards. Beyond the discussion of the achievements to date, future uses of TRIHYDE are briefly addressed.

## 2. Monitoring and Quantifying Gas Mixtures: Concepts and Calibration Strategies

Probably one of the most difficult and complex issues to resolve in quantitative analysis is that of validating measurement data, through absolute calibration of sample properties. In general, one may follow one of two main lines of action, namely

(A)To measure and compare the results from the sample under analysis to the data from known reference materials (standards); or(B)To analyze the sample using equipment, which is based on a metrological (certified) method.

Whilst analytical measurements correlated to certified reference materials (CRM) are preferred, in general, those are only available for a limited number of materials. In this context, note that reference materials that do not meet all the criteria to count as a CRM require additional evidence of metrological traceability (procedural routes are defined in Ref. [19]).

For mixtures of gases, which are the topic of this publication, various types of reference materials might be considered. These include (i) pure gases, characterized for chemical purity and/or trace impurities—evidenced normally by the purity labelling on the supplied gas bottle(s); (ii) standard gas mixtures, by and large prepared gravimetrically from pure substances—provided routinely by suppliers of specialty gases; or (iii) so-called matrix reference mixtures—characterized for the composition of specified major (matrix), minor and trace chemical constituents. Guidance on the terminology and definitions associated with such standards can be found in Ref. [20]

For the hydrogen isotopologues discussed in this work, only H_2_ and D_2_ are widely available with very high purity and certification; the heteronuclear species, HD, is less easy to source, but some specialty gas vendors offer it with about 96 mol%. Also, some recipes for the generation of reasonably pure HD have been published (see [21,22,23]).

When it comes to tritium, reference standards are not at all available at present; indeed, it was one of the purposes of this work to develop methodologies to do just that. The initial goal of the program was to achieve quantitative (cross) calibration of the composition of the tritium gas injected into the source of the KATRIN experiment, which then can be applied in the continuous monitoring of relative component concentrations. For this, in association with the calibration approach (B) defined above, a method suitable and proven for continuous in-line analysis and monitoring of tritiated gas samples was needed to characterize and quantify TLK-prepared samples to be used as reference standards. Raman spectroscopy was selected as the method of choice because of its universality, non-contact capabilities, and its proven track record for in-line gas composition monitoring in the KATRIN experiment [7].

The composition of any gas mixture can be extracted from Raman spectra using the relation between the (measured) signal of the Raman transition and the number of interacting molecules. The full expression for this relation is given by
(1)$$S_{\Delta \upsilon ,\Delta J}\left(J\prime \prime \right)=\underset{R_{\Delta \upsilon ,\Delta J}\left(J\prime \prime \right)}{\underset{︸}{k_{\tilde{\nu }}\cdot {\tilde{\nu }}_{L}\cdot {\tilde{\nu }}_{\Delta \upsilon ,\Delta J}^{3}\left(J\prime \prime \right)\cdot \Phi_{\Delta \upsilon ,\Delta J}\left(J\prime \prime \right)\cdot I_{L}\cdot \eta \left(\lambda_{\mathrm{Raman}}\right)}}\cdot N\left(J\prime \prime ,T\right).$$

Here, $S_{\Delta \upsilon ,\Delta J}\left(J\prime \prime \right)$ stands for the Raman signal for a transition from an initial vibrational | rotational state *υ*″ = 0|*J*″ to a final state *υ*′ = *υ*″ + 1| *J*′ = *J*″ − 2, *J*″, *J*″ + 2; these correspond to the O_1_(*J*″), Q_1_(*J*″) and S_1_(*J*″) Raman lines (note that in this work we only include the vibrational Δ*υ* = 1 transition in the evaluation). All dimensional proportionalities and numerical constants are collated into a single factor, $k_{\tilde{\nu }}$. The terms ${\tilde{\nu }}_{L}$ and ${\tilde{\nu }}_{\Delta \upsilon ,\Delta J}^{3}\left(J\prime \prime \right)$, as common in Raman spectroscopy in cm^−1^, are associated with the energy of the laser photon and the Raman-scattered photon. Note, the separation into laser and Raman transition energies is a consequence of our detection method in which photons are counted. The line strength function $\Phi_{\Delta \upsilon ,\Delta J}\left(J\prime \prime \right)$ incorporates the theoretical transition matrix elements for Raman scattering. $I_{L}$ is the intensity of the Raman excitation laser and $\eta \left(\lambda_{\mathrm{Raman}}\right)$ is a function associated with the wavelength-dependent response and sensitivity of the detection instrumentation. Finally, $N\left(J\prime \prime ,T\right)$ is the number density of the molecule under investigation in its initial state(s) *J*”; the partitioning into individual quantum states depends on the temperature *T*. As indicated in Equation (1), it is convenient to lump all parameters together into a response function, $R_{\Delta \upsilon ,\Delta J}\left(J\prime \prime \right)$.

Note that the molecular parameters in Expression (1) are all different for any species, here the six hydrogen isotopologues XY with X,Y ∈ {H,D,T}.

In general, several rotational states are thermally populated at room temperature. This means that in order to arrive at the actual concentration of a species XY, the number densities for the full rotational level population (Boltzmann distribution) need to be summed, i.e., $N_{\mathrm{XY}}=\sum_{J\prime \prime }N\left(J\prime \prime ,T\right)$. Note that for all measurements described in this work, the temperature was kept constant at T = 25 ± 1 °C. As a consequence of the *J*”-distribution the Raman line signals also need to be summed. One can approximate the total Q_1_-branch Raman signal as
(2)$$S_{\mathrm{XY}}=\sum_{J\prime \prime }S_{1,\Delta J=0}\left(J\prime \prime \right)=\sum_{J\prime \prime \;}\left(R_{1,\Delta J=0}\left(J\prime \prime \right)\cdot N\left(J\prime \prime ,T\right)\right)≅R_{\mathrm{XY}}\cdot \underset{J\prime \prime }{\sum }N\left(J\prime \prime ,T\right)=R_{\mathrm{XY}}\cdot N_{\mathrm{XY}},$$

Note that the approximation using a nearly constant “global” response factor $R_{\mathrm{XY}}$ in the right-hand part of Equation (2) is only valid for the Q_1_-branches (with Δ*J* = 0). This is because those lines are closely packed and none of the parameter values ${\tilde{\nu }}_{1,\Delta J=0}^{3}\left(J\prime \prime \right)$, $\Phi_{1,\Delta J=0}\left(J\prime \prime \right)$ and $\eta \left(\lambda_{\mathrm{Raman}}\right)$ change significantly, mostly less than 1% over the measurement interval. Note, also, that such global response factors are also assumed in the related Raman cross-sections, reported in the literature (see Ref. [24]). Finally, the $R_{\mathrm{XY}}$ values are different for each individual isotopologue and the associated Raman transition moments and are specific to the actual measurement instrumentation.

The absolute quantities in Equations (1) and (2) can be transposed into relative entities by normalizing the respective (relative) Raman signal intensities to the integral signal intensities of all components present in the mixture, i.e.,
(3)$$S_{\mathrm{XY},\mathrm{rel}}=S_{\mathrm{XY}}/\left(\sum_{X,Y}^{}S_{\mathrm{XY}}\right)≅\left(R_{\mathrm{XY}}\cdot N_{\mathrm{XY}}\right)/\left(\sum_{X,Y}^{}S_{\mathrm{XY}}\right),\;$$
with the summation over all isotopologues XY with X,Y ∈ {H,D,T}; note that not all may be present in particular samples. The relative-value representation in Equation (3) has the advantage that inevitable fluctuations in laser intensity $I_{L}$ cancel out in the ratios.

An example for spectral Raman data utilized in the application of Equations (1)–(3) for compositional analysis is shown in Figure 1 for a mixture of T_2_-D_2_-H_2_, initially in the nominal equi-volumetric ratio 1:1:1.

The individual response factors $R_{\mathrm{XY}}$ may be decomposed into two contributions, namely $R_{\mathrm{XY},\exp}\equiv C\cdot \eta \left(\lambda_{Q_{1}\left(\mathrm{XY}\right)}\right)$ with $C=k_{\tilde{\nu }}\cdot {\tilde{\nu }}_{L}\cdot I_{L}$ (constant during a spectral acquisition)—the average spectral sensitivity of the Raman system over the Q_1_-branch wavelength interval (note that for the related spectral intervals of ~4 nm the sensitivity only varies by 1.2–2.2% and, thus, may be approximated by its mean value); and $R_{\mathrm{XY},\mathrm{theory}}$—the theoretical Raman intensity for the Q_1_-branch of the isotopologue XY, including the quantum state-dependent terms for the transition energies ${\tilde{\nu }}_{\Delta \upsilon ,\Delta J}^{3}\left(J\prime \prime \right)$ and the transition probabilities $\Phi_{1,\Delta J=0}\left(J\prime \prime \right)$. This yields.
(4)$$R_{\mathrm{XY}}=R_{\mathrm{XY},\exp}\cdot R_{\mathrm{XY},\mathrm{theory}}.$$

This procedure will be discussed in more depth in Section 4.1. The main inference from Equation (4) is that one is now able to follow two complementary signal calibration strategies:(I)Use accurate mixed gas samples of hydrogen isotopologues to determine the individual global response factors $R_{\mathrm{XY}}$; essentially, this is equivalent to the procedure defined in the calibration method (A) at the beginning of this section; and(II)Measure the instrumentation-specific spectral sensitivities and combine that with the theoretical Q_1_-branch intensities for the isotopologues; essentially, this constitutes calibration method (B) at the beginning of this section.

Note that in all analytical measurements discussed here, only the Q_1_-branch summation/integral are utilized, and not individual rotational Raman lines, as is common in compositional analysis based on Raman spectra. A few remarks should be made in respect to the use of the two methodologies and cross-validation based upon them.

Firstly, in approach (I), and the underlying relations in Equations (3) and (4), the essential point is that multiple-component gas samples need to be used so that the numerator and denominator in Equation (3) are different.

Secondly, the use of relative (Equation (3)) rather than absolute (Equation (2)) values for the Raman signal response guarantees that the method is, by and large, insensitive to (unavoidable) fluctuations and drifts in the excitation laser intensity $I_{L}$.

Thirdly, in approach (II), one relies on the availability of (or access to) spectral sensitivity data $\eta \left(\lambda_{\mathrm{Raman}}\right)$ of the measurement setup and the theoretical Raman transition line strengths $\Phi_{1,\Delta J=0}\left(J\prime \prime \right)$. In our work, the spectral sensitivity function was obtained by using a NIST-certified luminous standard, placed for calibration at the exact location of the Raman excitation (see Ref. [25] and Section 3.3 for details). The theoretical line strengths depend on the molecule specific polarization tensor invariants; these were calculated specifically for the excitation wavelength λ = 532 nm [26]. Note that this calibration strategy was verified using accurate gas samples prepared in the predecessor HYDE experiment, albeit only for the stable isotopologues H_2_, HD, and D_2_ [16,27].

Thus, based on this discussion of measurement and calibration methodologies, the key objectives of the TRIHYDE gas mixing experiment may be summarized as follows:Enable reference gas sample production of up to all six hydrogen isotopologues, in chemical equilibrium;Achieve improved manometric accuracy for the sample mixtures, in comparison to the predecessor HYDE;Verify the theoretical intensities for tritiated hydrogen isotopologues; andReduce the calibration uncertainties relevant for precise compositional analysis of gas mixtures with tritium content, based on laser Raman spectroscopy.

These key aspects are outlined in the sections below, together with relevant technical details and selected application examples of the procedure in quantitative analysis and calibration.

## 3. System Setup

The manufacture of non-hazardous reference gas samples and provision of gas mixtures for specialist applications are straightforward, in principle, and are provided as a service by most suppliers of specialty gases. This is rather more complicated in the case that hazardous, toxic, or radioactive species are involved, which requires rare specialist facilities.

### 3.1. Experimental Challenges

For the technical realization of a H_2_-D_2_-T_2_ loop, the many challenges for tritium operation, as identified in Ref. [16], were addressed during the conception, construction, and operation of the TRIHYDE facility. These include

▪The possibility to study and exploit radio-chemical reactions, such as β-induced self-equilibration;▪The ability to concurrently use several analytical methods to characterize and/or monitor the gas mixture, as well as to verify—in situ—the initial T_2_ purity;▪The ability or necessity to remove non-hydrogen contaminants (such as ^3^He, the byproduct of the tritium decay) from the T_2_ gas sample; and▪The need to comply with all statuary safety requirements for tritium operation.

Therefore, much more stringent design guidelines had to be followed, as was done for the predecessor experiment, HYDE [16], which was set up for the mixing of non-radioactive hydrogen isotopologues.

In order to meet all requirements of the regulatory framework, and to allow for efficient gas processing and sample analysis, the TRIHYDE experimental setup comprises two main functional subsystems. These are (i) the processing loop (P-Loop) for gas handling and interface to the gas processing infrastructure of TLK and (ii) the analysis loop (A-Loop), which is used to prepare and analyze the gas samples.

The integration of TRIHYDE into the gas processing infrastructure of TLK is illustrated schematically in Figure 2. For a T_2_ mixing run, the P-Loop is supplied with a high-purity T_2_ sample (normally > 98%) from the tritium transfer system (TTS), which is the central distribution and accountancy system of TLK. This gas is then transferred to the A-Loop, where it is mixed with other gases to produce the desired gas mixtures. After analysis, the gas is pumped back into the P-Loop, from where it is passed on to either (i) the so-called CAPER facility, in which any gas sample arising from experiments using tritium is detritiated [28,29], or (ii) to the isotope separation system (ISS) [30], depending on composition and activity level.

In order to ensure that no hazardous atmosphere forms, and to provide a secondary enclosure to the primary tritium system, TRIHYDE is constructed within a dedicated glovebox environment, providing an inert N_2_ atmosphere. Unavoidable tritium contamination of the glovebox atmosphere, due to permeation through the stainless steel and during (dis)connecting sample cylinders, is continuously removed to the tritium retention system (TRS).

For all operational components—like valves, pumps, sensors, etc.—the wetted materials are made from stainless steel and, in rare cases, from halogen-free polymers. Connections between components use stainless steel piping and VCR fittings; this ensures single-connection He leak rates below 10^−9^ mbar·*ℓ*·s^−1^ and an integral leak rate smaller than 10^−8^ mbar·*ℓ*·s^−1^.

### 3.2. The Analysis/Measurement Sub-System of TRIHYDE—The A-Loop

A detailed schematic of the A-Loop is shown in Figure 3. The molar fractions of the gas samples are determined using a manometric method, based on accurate measurements of the mixing vessel volumes and gas pressure (see Ref. [31]). For the pressures typically used for sample production (10^−3^ mbar to 800 mbar), the ideal gas law is applicable. Both identical mixing vessels are built from standard CF vacuum components and they have an inner volume of *V* = 810 (± 2) cm^3^.

In order to provide accurate and species-independent pressure readings, each mixing vessel is equipped with two capacitive pressure gauges, with full scales of 1000 mbar and 2.6 mbar, respectively, and accuracy of 0.15% of the pressure reading. Note that in comparison to the mixing vessels and connecting tubing of HYDE, TRIHYDE is designed to minimize “dead volumes” with no gas mixing and to provide a straight flow geometry through the vessels.

Both mixing vessels can be filled either with T_2_, supplied via the P-Loop, or non-radioactive gases introduced from gas bottles through a dedicated filter and pressure regulation stage.

For the measurements presented in this work, H_2_ and D_2_ were used for mixed-sample production, as well as ^4^He for purging of the loop system. For future campaigns, it is possible to expand the gas supply capabilities, to allow for admixture of other gas species of interest, e.g., Ar and/or N_2_.

In general, for sample production, the mixing vessels are filled with the homonuclear hydrogen isotopologues (H_2_, D_2_, or T_2_). In order to form the heteronuclear isotopologues (HD, HT, or DT) on a reasonable timescale (i.e., minutes to hours, instead of many weeks), the mixed gases can be circulated through a catalyst bed (e.g., Pt−Al_2_O_3_). If necessary, the catalyst and the mixing vessels can be heated up to 200 °C for catalysis acceleration or improved outgassing. Note that, for gas mixtures containing T_2_, the process of β-induced self-equilibration, as described in Section 4.1, leads to self-equilibration of the gas samples, but without the need for a catalyst. Thus, a potential shift in the isotopic ratio—introduced by the catalyst material—can be avoided.

For the most part, the general concept of the A-Loop is based on its predecessor experiment HYDE [16], incorporating the following improvements: (i) pressure diagnostic over a wider pressure regime in the mixing vessels, with increased accuracy; (ii) reduced mixing vessel volume for more economical gas usage; and (iii) additional temperature and pressure sensors within the loop. In addition, the component placement and routing have been optimized to minimize piping lengths and to avoid the aforementioned dead volumes, while at the same time offering easy access for maintenance and operation inside the glovebox environment. Along the loop, several sampling ports are installed, which allow one (i) to fill gas samples into sample cylinders, to be used in other experiments, or (ii) to accept gas samples from other experiments, to make use of the broad range of analysis tools installed in the A-Loop. Key parameters for the TRIHYDE facility are collated in Table 1.

### 3.3. The Raman Analysis Instrumentation

The central analysis instrumentation of the TRIHYDE facility is a laser Raman (LARA) system; it is one of several setups in use at TLK, whose general layout and performance are described in detail in Ref. [7].

After passage through a range of steering and polarization-cleaning optics, the excitation laser beam is focused into the in-line LARA gas cell, which is located within the A-loop glove box (see the previous Section 3.2). The Raman scattered light is collected using a standard collection geometry of 90° to the laser excitation direction, and is then guided to the spectrometer, with CCD array detector, via optical fiber coupling. A few details about the equipment are summarized below; key properties of the LARA monitoring system are collated in Table 2.
**The laser excitation source**. Due to the direct proportionality of laser power and scattered Raman light intensity (see Equation (1) in Section 2), and further response reduction due to below-atmospheric pressures inside the LARA cell, high laser power is required to obtain Raman signals with good signal-to-noise ratio. Thus, a high power, temperature-stabilized DPSS laser (λ_L_ = 532 nm) is used, with power fluctuations and drifts at the level of ~10^−3^.**The Raman cell**. The Raman cell is of the same type as that used in the KATRIN experiment [7], based on the original design concept described in Ref. [32]; it is mounted inline within the second-containment glove box.**The light collection optics and fiber bundle**. An assembly of two achromatic doublets is used to image the observable Raman excitation volume onto a matched “slit-to-slit” fiber bundle. A steep-edge long-pass filter is used to suppress light originating from Rayleigh scattering, and from laser light reflections within the Raman gas cell.**The spectrometer**. A common combination of a standard Czerny–Turner spectrograph and a back-illuminated CCD array detector is used for recording the Raman spectra. The spectrograph is set to comfortably cover the all-important Raman bands of all six hydrogen isotopologues, maintaining sufficient spectral resolution to separate the individual Q_1_-branch signals, which are later integrated for the evaluation of relative molecular concentrations (for the procedure, see Section 4.3).


Note that for absolute, quantitative light intensity calibration of the complete Raman system a NIST SRM2242 Raman standard [33] can be inserted, at the exact location of the Raman gas cell; for a description of the calibration procedure, the reader is referred to References [7,25]. In brief, the actually measured SRM2242 fluorescence spectrum is normalized to the certified spectrum provided by NIST (the spectrum is parameterized by a polynomial). Note that the increase in uncertainty beyond ~660 nm, observed for the $\eta \left(\lambda \right)$ data shown in Figure 1, is associated with the diminishing accuracy of the certified spectral luminescence data, provided by NIST, towards the largest Raman shift values [33]. Wavelength calibration is carried out as is common, i.e., using a Ne hollow-cathode lamp.

### 3.4. Ancillary Analysis Equipment

When circulating gases in the A-loop, their composition and activity are continuously monitored using a chain of in-line tools (see Figure 4). In addition to the laser Raman spectroscopy (LARA) system (for composition monitoring, as described in Section 3.3 above), these comprise: (i) a binary gas analyzer (BGA) for composition monitoring (see e.g., [34]), (ii) a beta-induced X-ray spectrometry device (BIXS) for activity monitoring (see e.g., [35]), and (iii) an ionization chamber (IC) for activity monitoring (see [36]). These three additional in-line methods have the common task to improve real-time composition monitoring of the samples and radiation safety aspects; they complement the data measured by the main Raman (LARA) spectral analysis system. All measurements are synchronized using NTP-derived time stamps (network time protocol). Below, we briefly summarize the key aspects of these ancillary analysis methods, as utilized in TRIHYDE.
**Binary Gas Analyzer**. A BGA can be used to determine the concentration of a gas component in a binary mixture by measuring the speed of sound and the temperature. Using known (tabulated) speed-of-sound data, the composition of (binary) gas mixtures can be determined with an accuracy of the order of 10^−3^ [37,38]. BGAs are less suitable in situations when more than two gas components are encountered or if the gas properties required in the evaluation equations are not known a priori (as can be the case for tritium-substituted species). In our TRIHYDE loop, a BGA is installed in-line. The device can detect differences in relative concentration shift of the order 0.1%, but needs >350 mbar to operate reliably. Thus, it is predominately used to confirm initial gas purities and to verify the homogenization time constants of the calibration gas sample.**Beta-induced X-ray spectrometry**. The activity of the circulated gas is continuously monitored using a BIXS measurement cell, based on the design described in Ref. [35]. The bremsstrahlung X-ray spectrum, produced by tritium β-electrons stopped in a thin gold (Au) layer, is a direct indicator for the gas activity level (for quantification, known pressure correction need to be applied). Besides applications for in-line process control [39], this sampling technique is successfully deployed for activity monitoring of the tritium source (WGTS) of the KATRIN experiment [40].**Ionization chamber**. At the TLK, ionization chambers—based on stainless steel cross-pieces—have been used for more than 20 years to monitor the activity of tritiated gases [41,42]; their design and performance have been optimized over the years (see Ref. [36]). Such a second-generation ionization chamber has been incorporated into the TRIHYDE analysis loop, continuously monitoring the activity of the circulating gas.**Rest gas analysis by mass spectrometry**. For the identification of trace amounts of non-hydrogen components in the loop, a quadrupole mass spectrometer is used as a residual gas analyzer (RGA). Due to its maximum operating pressure of ~1 × 10^−3^ mbar, which is substantially lower than the typical mixing pressures and thus requires active pumping, the RGA is placed off-line to the main mixing loop. Since the limited mass resolution of our current device does not allow for the separation of sub-amu fractions, unambiguous separate quantification of hydrogen isotopologues is not possible. Therefore, at present, the RGA is mainly used to ascertain the presence of gas species above 9 amu, in order to evaluate the formation of impurities during gas circulation.


### 3.5. The Processing Sub-System of TRIHYDE—The P-Loop

The main task of the P-Loop (its schematic structure is shown in Figure 4) is to supply highly pure tritium gas to the A-Loop for sample production and to process the mixed gas samples after use. In this way, sample production and operation of the A-Loop can be realized without the need for continuous gas transfers to and from the tritium processing infrastructure of TLK.

Its operation principle is as follows. For each measurement campaign, the tritium gas batch from the TTS is transferred into a T_2_-buffer vessel. In order to ensure high T_2_ purity, the initial TTS gas sample can be stripped of ^3^He, which is naturally present due to tritium β-decay. For this, a palladium-silver (PdAg) filter is used (for details on operation, see Refs. [43,44]). From the T_2_-buffer vessel, the gas is expanded into one of the mixing vessels of the A-Loop for analysis and the subsequent experiment.

After each sample is prepared and analyzed, and an actual experiment has been completed, the used “waste” gas is collected from the A-Loop into an extraction buffer vessel. In order to sufficiently remove residual gas from the A-Loop, which might taint subsequent gas samples, the P-Loop includes three pumps in series (see Figure 4). With this pump combination, pressures of the order <10^−5^ mbar can be reached in the complete TRIHYDE system.

Once the waste buffer vessel pressure limit is reached, its content is transferred to the TLK gas processing infrastructure. In the case that only hydrogen isotopes are present, the gas is transferred to the ISS [30] for immediate isotope separation. In the case that tritiated mixtures with unknown composition (including reaction products generated during circulation) need to be treated, the gas is passed to the CAPER cleanup facility [28]. If low activity mixtures with trace amounts of tritium are present, these are transferred to the TRS for processing [45]. Note that the same permeator that was used for cleaning of the initial gas(es) can also be utilized to pre-process the waste gases from the A-Loop, allowing for a reduction in workload for the TLK processing infrastructure.

It is noteworthy that the separation of sample production/analysis (A-Loop) and gas processing (P-Loop) is beneficial. Future system expansions can conveniently be installed in the same secondary enclose as the current TRIHYDE mixing experiment, using the P-Loop as the interface to the TLK infrastructure.

## 4. Measurement Concepts

In contrast to calibration samples containing only stable hydrogen molecules—in our case that would be the isotopologues H_2_, HD and D_2_—the preparation procedure and analysis for tritiated mixtures must consider additional effects. These include variable T_2_ supply gas composition (differences in batch purity), self-equilibration, and unavoidable formation of impurities as a consequence of tritium β-decay (^3^ He decay product and induced chemical reaction chains). This means that a rigorous procedure for sample preparation and handling is required to ensure consistent and reproducible sample composition. Below, we outline how such samples are prepared, how they are characterized, and how data are recorded and evaluated to quantify their composition.

### 4.1. Methodology

The general strategy of TRIHYDE comprises two key aspects, namely (i) to produce gas samples, which may contain any combination of all six hydrogen isotopologues, with well-defined composition, and (ii) to extract calibration factors from appropriate measurement series for future characterization and analysis by comparing measured LARA signal intensity to “actual” gas compositions (see Equation (3) above). This then gives access to both absolute calibration factors, $R_{\mathrm{XY}}$, and theoretical intensities, deduced from $R_{\mathrm{XY},\mathrm{theory}}$, depending on whether the spectral sensitivity correction is applied to the measured Raman signals (see Section 2 and Section 4.3).

During sample production, two stages in the evolution of the gas composition are distinguished. The “binary” sample at the beginning of the process, which predominantly consists of two homonuclear isotopologues, such as e.g., H_2_ and T_2_, with only trace impurities of other isotopologues. The “tertiary” sample at the end of the preparation/equilibration process is made up from two homonuclear and the associated heteronuclear isotopologues. Analyzing both binary and tertiary stages of a sample in conjunction has multiple advantages:▪The binary sample can be produced with high accuracy and its actual composition is not influenced yet by chemical equilibrium constants.▪The extraction of the calibration factors from a two-component mixture is simpler in comparison to a three-component mixture and, thus, carries less measurement uncertainty.▪The tertiary sample gives access to the heteronuclear isotopologues HT and D, which normally cannot be produced in ultra-pure form and are difficult to store in isolation for practical periods of time. This is due to the inevitable self-equilibration described by Equation (5).

In order to analyze well-defined binary/tertiary mixtures (e.g., H_2_-T_2_ and H_2_-HT-T_2_), the initial binary calibration sample needs to proceed towards chemical equilibrium concentrations, according to
X_2_ + Y_2_ ⇌ 2·XY with X, Y ∈ {H, D, T}.(5)

These isotope exchange reactions (in the gas phase) have been studied extensively for H_2_ and D_2_; however, for the cases where radioactive isotopologues are involved, only a small number of experimental results have been published (see references [46,47,48,49]). Said exchange reactions in the gas phase can be organized into distinct groups [50].

The first group of reactions is that of “direct saturated molecule interactions”. For a mixture of inactive molecules—here H_2_ and D_2_—this is the only possible exchange reaction. Due to the high activation energy and low probability, these direct reactions exhibit inherently long evolution times to the final, chemical equilibrium [51]. These can be shortened by use of a catalyst environment which, however, may introduce additional uncertainties and potential, undesired effects on sample composition.

In tritiated mixtures, the tritium β-decay generates various ion radicals, which bring about subsequent radiochemical reaction chains; these greatly accelerate the self-equilibration process [52]. In principle, it is possible to exclusively rely on these reactions to guide the gas sample into chemical equilibrium within a practical timescale. Thus, this approach avoids any uncertainties introduced through the reaction sequences associated with a catalyst material. In order to illustrate this type of self-equilibration process, the measured evolution of isotopologue concentrations for a gas sample of D_2_-T_2_ (with an initial ratio of 40:60) is shown in Figure 5, while the mixture is circulating in the A-Loop. The progress of equilibration reaction for the main constituents is clearly visible.

Note that for the unavoidable HT and HD impurities, additional, superimposed equilibration processes are observed (due to gas–wall reactions), however on a much longer time scale (of the order ~10^−5^ h^−1^).

The molar fractions of a tertiary sample in chemical equilibrium evolving from T_2_ plus X_2_, with X ∈ {H, D}, are given by
(6)$$k\mathrm{eq}\left(T\right)=\frac{{\left[\mathrm{XT}\right]}^{2}}{\left[X_{2}]\cdot [T_{2}\right]}.$$

The (temperature-dependent) chemical equilibrium constants, $k\mathrm{eq}\left(T\right)$, for each combination of T and X used in this work were derived from the ro-vibrational energy levels reported in [53], which agree with older published values [46] to within 0.5% (at room temperature). For a detailed description of how the equilibrated molar fractions—based on the initial, binary fractions—were calculated, see references [16,54].

### 4.2. Sample Preparation and Handling

For all calibration campaigns presented here, which used either H_2_-T_2_ or D_2_-T_2_ starting mixtures, a single batch of T_2_ was transferred from the TTS to the P-Loop. From there, each individual sample preparation process consisted of three main steps:(1)The verification of initial gas purity;(2)The production of a binary sample; and(3)The production of the tertiary (equilibrated) sample.

At the beginning of each sample preparation sequence, the complete A-Loop was evacuated to pressures <1 × 10^−3^ mbar. For typical gas samples with partial pressures of normally >50 mbar, this ensures that background impurities are reduced to negligibly low levels. In addition, the composition of the background gas is checked each time using the RGA.

Due to possible leakage through safety valves and long connection pipes from the gas cabinet, which supplies the inactive gases (here, H_2_ or D_2_) to the system, the respective gas filters were flushed with a flowrate >200 sccm for about 10 min. This procedure proved to be very effective to eliminate trace amounts of N_2_ and other impurities to below the detection level of the RGA (of the order 10^−9^ mbar) and LARA. Subsequent to the flushing, the desired gas was injected and its purity verified, using LARA, BGA, and RGA measurements in combination. If any noticeable impurities were detected, the sample was discarded and the filters flushed repeatedly until acceptable gas purity was reached. Finally, one of the mixing vessels was filled (in general to values in the range 50–450 mbar), and the whole loop—apart from the just filled mixing vessel—was again evacuated to <1 × 10^−3^ mbar.

The filling procedure for tritium was analogous to that just described for the stable isotopologues, with the difference that the T_2_ gas was expanded from the P-Loop and, thus, filter flushing was not applicable. Again, the initial gas purity was recorded using LARA, BGA, and RGA (see Section 5.1), and after termination of the filling process, the remaining loop tubing was evacuated to <1 × 10^−3^ mbar.

For the subsequent mixing process, all valves along the mixing loop were opened and the gas sample continuously circulated using a metal bellows pump. On average, it took about 5 min for the sample to fully homogenize, verifiable by the fact that the maxima of both the T_2_- and Q_2_-signals had stabilized.

It has to be noted that, upon gas contact after starting the mixing circulation, the β-induced self-equilibration process commences immediately. Detailed studies of the impact of process parameters on the self-equilibration reaction kinetics and their temporal evolution are ongoing; however, for the scope of this work, it was shown that, for a combined pressure in the mixing vessels of the order of 500 mbar, the chemical equilibrium state was normally reached within ~72 h. Throughout the gas circulation period up to equilibration, LARA spectra were continuously recorded with an acquisition time of 60 s.

In order to avoid gas extraction during circulation, a single, final RGA analysis was performed after termination of the mixing/equilibration process. Thereafter, the sample was discarded and pumped to the P-Loop for further processing in the TLK infrastructure.

Overall, the complete sample preparation process took about half a day; thereafter, the time needed to reach chemical equilibrium via self-equilibration was about three days, in general.

### 4.3. Data Evaluation

An in-depth description of the spectrum processing can be found in references [7,55]; thus, only a brief summary of the major processing steps is given here. In general, prior to the extraction of the concentration data, the following procedures were applied to each individual LARA acquisition:▪On-chip binning of the 400 lines of the *Pixis*-detector into 20 bins of 20 lines each;▪Dead pixel and cosmic ray removal;▪Baseline correction using the Savitzky–Golay Coupled Advanced Rolling-Circle Filter (SCARF) method; and▪Correction for the spectral response of the LARA system (based on calibration with the SRM2242 Raman standard).

All these form part of our bespoke, LabVIEW-based analysis software suite for the evaluation of Raman spectra, *LARAsoft* ([55] and references therein)

In order to derive the concentration data from the measured Raman spectra, the signal intensity *S_x_* for each isotopologue is integrated over the respective Q_1_-branch wavelength interval.

These intervals were determined using the function *signal.peak_prominences* of the Python library *SciPy* [56], applied to Raman spectra of an equilibrated tertiary test gas sample, with equal component content at the start of the equilibration process. This resulted in similarly intense Q_1_-branch signals for any of the six hydrogen isotopologues.

The “prominence of a peak”, *P*, describes how much a peak stands out from the baseline within a specified interval; it is defined as the vertical distance between the peak maximum and its lowest contour line. Here, this particular procedure was selected as it is suitable to derive the overall peak width at the baseline. This results in reproducible intervals and is not affected by the actual shape of individual Q_1_-branches. After initial trials, a value of *p* = 0.98 was chosen; this constitutes a trade-off between maximum signal strength and robustness against baseline noise, and independence of the initial search intervals. Note that this *p*-value is consistent with the notion that all levels with a relative thermal population and associated relative line strength larger than about 1% are included in the integration (see Figure 6), reflecting the full thermal population distribution.

At the bottom of Figure 6, another aspect relevant for the evaluation of isotopologue concentrations is highlighted, namely the overlap between the Q_1_-branch of one isotopologue with S_1_- and/or O_1_-lines of another one. In this context, note that the line strengths of S_1_- and/or O_1_-branch lines are at least 20-fold weaker than those for the Q_1_-branch lines.

While such overlaps may not constitute a major problem in the case that the concentrations of the involved isotopologues are similar, such overlaps need to be disentangled in mixtures with very different concentrations of the contributing isotopologues. For example, in the gas mixtures used in the KATRIN experiment, with a T_2_ content of normally ≥ 98%, the overlap T_2_ S_1_(3) ↔ DT Q_1_(*J*″) would yield a concentration erroneous by up to 30% if not treated appropriately, as outlined in Ref. [7]. In the evaluation of the spectra from our gas mixtures, with widely varying concentration ratios, this separation procedure was applied as soon as the expected contribution surpassed the 1–2% level.

Similar to the observations presented in Ref. [52], and demonstrated in our own preliminary studies [54], the temporal evolution of the molar fraction, $y_{\mathrm{XY}}$, for each isotopologue XY in a mixture of only two hydrogen isotopes can be described by
(7)$$y_{\mathrm{XY}}\left(t\right)=A_{\mathrm{XY}}\cdot \exp\left(-t/\tau_{\mathrm{XY}}\right)+C_{\mathrm{XY}},$$
with characteristic time constant, $\tau_{\mathrm{XY}}$, and equilibrium concentration ratio, $C_{\mathrm{XY}}$. This function is fitted individually for each isotopologue in the actual calibration sample. The resulting fitting parameters are used to derive the relative signal strengths for the binary (t=0) and tertiary (t→∞) samples.

For extracting the relative LARA calibration factors, $R_{\mathrm{XY}}$, Equation (2) must be solved simultaneously for all constituent isotopologues. The molar fractions, $y_{\mathrm{XY}},$ are derived from the filling parameter of the mixing vessels and the relative signal strengths, $S_{\mathrm{XY}}$, from the aforementioned integration. The resulting calibration curves (see Section 5) are then derived by simultaneously minimizing the residuals for the two (binary) or all three (tertiary) isotopologues, using the Levenberg–Marquardt algorithm implemented in the Python package *lmfit* [57]. The advantage of using both the binary and tertiary sample data is that it greatly reduces the fitting complexity and minimizes the final uncertainty.

## 5. Analytical Measurements

In order to successfully prepare and use any precision gas mixture for a particular application, several analysis/verification steps are required: (1) verification of the nominal initial gas composition, (2) verification of all calibration procedures using only the stable isotopologues, and (3) deriving calibration factors from mixtures containing tritiated isotopologues. Only then is it possible to provide reproducible, well-characterized samples reference gas mixtures. All aspects are outlined in the sections below for a selected range of sample mixtures.

### 5.1. Verification of Initial Component Purities

All samples were prepared according to the procedure and parameters as described in Section 4.2. During all processing steps, the gas was fully thermalized at ambient temperature (25 ± 1 °C). Within these limits, the respective chemical equilibrium constants *k*_eq_(*T*) (see Equation (6)) only vary by <0.05% and, thus, have only a negligible effect on the equilibrium molar fractions.

When dealing with tritiated samples, the key difference is the variable initial batch purity due to (i) unavoidable presence of HT and DT in the initial T_2_ “feed” gas and (ii) formation of impurities, e.g., tritiated methane, associated with β-induced reaction chains. Both issues were addressed and were well controlled for subsequent preparation of mixed samples.

For each tritium campaign, a single batch of T_2_ was stored in the P-Loop buffer vessel, and its actual gas composition was measured prior to use. It was shown that each batch contained mainly T_2_ (>97%), with varying amounts of DT and HT (0.5–2.5%) as impurities. Although each initial batch showed a slightly different HT to DT ratio, the gas composition used within each campaign remained constant for all practical purposes (changes over campaign time < 0.1%). A detailed description of the composition measurement of the tritium samples is given in Ref. [49].

The composition values for the tritium batches and the stable isotopologues are collated in Table 3. The values for the latter were based on the gas purities stated by the supplier, which were verified using LARA and RGA.

The formation of impurities in gas samples with T_2_ content is a well-documented, albeit not fully understood, phenomenon. In order to assess its impact on the composition on the later gas mixtures, the system was filled with a “pure” T_2_ sample (loop pressure ~30 mbar), and the gas composition was continuously monitored. During >1500 h of gas circulation (bypassing the PdAg-filter) at ambient temperature, the following observations were made:(1)In the measurement data obtained with LARA and RGA, an increase in signals associated with tritium-substituted methane was detectable, with an approximate relative formation rate of ~5 × 10^−6^ h^−1^. No formation of CO and CO_2_ was observed.(2)A constant decrease in T_2_ content and increase in HT concentration was detected, at a rate of ~7 × 10^−6^ h^−1^, which can be attributed to β-induced reactions with the available hydrogen reservoir in/at the stainless steel walls, similar to the effects described in Ref. [58].

As a consequence, the observed low rates were of no concern for the typical time scale of ~72 h to reach chemical equilibrium. For verification of this, all gas mixtures were checked for impurities at the beginning and end of each individual preparation and mixing run, using RGA analysis.

### 5.2. Measurement Results Obtained for Inactive Isotope Mixtures

In order to evaluate the system’s performance and to understand the impact T_2_ might have on the sample preparation procedure, ten H_2_−D_2_ samples with initial molar fractions from 5:95 to 95:5 were prepared, according to the procedure outlined in Section 4.1. For each calibration sample, the initial molar fraction of a particular isotopologue was calculated from the inlet pressure, vessel volume, and gas purities. The Raman spectra were treated following the procedure outlined in Section 4.3. However, the exponential fit for data from gas mixtures with tritium content (see Equation (7)) is not applicable for a mixture only consisting of stable isotopologues. Thus, the Raman signal strength was obtained by averaging the signals over the total circulation time of ~20 min for each sample.

The analysis principle to extract the calibration factors *R*_XY_ has been introduced in Section 4.3, which is based on the procedure described in detail in Ref. [16]. Note that the absolute value of an individual calibration factor $R_{\mathrm{XY}}$ is not of physical importance for relative quantification, but the ratio between two calibration factors is, i.e.,
(8)$$R_{\mathrm{VW}}/R_{\mathrm{XY}}\equiv R_{\mathrm{VW},\mathrm{XY}},$$
with V, W, X, Y ∈ {H, D, T}; here VW indicates the isotopologue against which the component XY is referenced. Ratios as in Equation (8) are used in the cross-calibration application discussed in Section 5.5.

For the H_2_-D_2_ campaign described here, the ratio $R_{H_{2},D_{2}}$= 1.2063(80) was derived. Note that this particular result is only valid for the setup configuration used during this measurement campaign, and not globally. However, the low uncertainty value of ~0.7% gives an indication about the relative composition accuracy, which can be achieved in TRIHYDE.

Note also that, in principle, the heteronuclear isotopologue HD can be generated in H_2_-D_2_ mixtures when circulating the mixture via the catalyst pathway (see Figure 3). However, such measurements were not included here. This is because the catalytic reaction environment introduces difficult-to-handle effects, associated with the temperature-dependent catalytic exchange reaction rates and isotope-specific adsorption probabilities on the catalyst surface. While these effects can be taken into account (see Ref. [16]), a link to the campaigns with T_2_ and the associated β-induced equilibration is rather difficult to establish.

### 5.3. Measurement Results Obtained for Tritiated Isotopologue Mixtures

For mixtures starting with T_2_ gas in the mixture, three measurement campaigns were performed: (I) H_2_-T_2_ at 100 mbar; (II) H_2_-T_2_ at 500 mbar, and (III) D_2_-T_2_ at 500 mbar. Note that the designated pressure values represent the combined pressure in both mixing vessels before mixing; hence, the resulting circulation pressure is lower. For each of the three campaigns, sets of samples were prepared and analyzed, with initial molar fractions ranging from 10:90 to 90:10. Samples with initial molar fractions of 50:50 were prepared multiple times (they exhibited the largest overall uncertainty) and, thus, give insight on the reproducibility of the procedure to generate equilibrated samples. The calibration curves derived from the sample sets (II) and (III) are shown in Figure 7.

For every sample, the initial relative isotopologue fractions for the “binary” sample were derived—based on the ideal gas law—using the initial filling pressure, mixing vessel volumes and gas purity. In order to acquire the molar fractions of the “tertiary” samples, Equation (7) was used with the initial molar fractions and a fixed value for the equilibrium constant of *k*_eq_(*T* = 298 K) = 2.574 and *k*_eq_(*T* = 298 K) = 3.811 for HT and DT, respectively. The relative Raman signals for “binary” and “tertiary” sample composition, i.e., at the beginning of the mixing process and after equilibration, respectively, were derived using the model function given in Equation (7); the related exponential fit incorporated all data values over the full equilibration circulation period of ~72 h, as illustrated in Figure 5 above.

The shown calibration curves were derived by simultaneously solving Equation (3) for the binary/tertiary combinations XX-YY and XX-XY-YY, respectively. Note that the non-trivial calibration factor introduces a detectable shift from the $R_{\mathrm{XY}}$ = 1 relation between initial molar fractions and the measured (normalized) Raman signals. The deviation of the data point sequence from this form is largely associated with the difference in the spectral sensitivity for the various isotopologues (see spectral calibration graph in Figure 1). Inspection of the H_2_-T_2_ and D_2_-T_2_ data plots reveals distinct curvatures, but because of the narrower spectral separation of the latter two isotopologues it is less pronounced in the D_2_-T_2_ graph. As an example, the straight-line case $R_{\mathrm{XY}}$ = 1 is illustrated for the binary mixture of H_2_-T_2_ (top-left panel in Figure 7). As mentioned in the previous section, the conversion from the binary into the tertiary sample by circulating the gas sample over a catalytic material may induce shifts in the relative atomic isotope ratio. By solely relying on the process of self-equilibration, this difficult-to-control catalyst influence is eliminated; no change in the relative atomic ratio in any of our equilibrated sample mixtures was observed. Thus, the initial fractions from the binary sample can be used to derive the composition in chemical equilibrium. An overview of extracted relative calibration factors can be found in Ref. [54].

It should be noted that during campaign (I), which was performed at 100 mbar, no fully equilibrated tertiary sample was produced. Due to the fact that the equilibration time constants are strongly dependent on pressure, the 72 h of circulation were insufficient to reach chemical equilibrium. On the upside, the evolution towards chemical equilibrium was slow enough to assume a constant gas composition during the data averaging time interval of 10–20 min. This yielded lower uncertainties compared to the samples from campaign (II). In particular, this proved to be beneficial at the beginning of the mixing process when the rate of change was largest.

The higher pressure of 500 mbar used in the campaigns (II) and (III) provided a higher signal-to-noise-ratio in the Raman signals. However, the progressing self-equilibration and the need to describe the process via a model-function offset this advantage and may have even led to slight alterations in the calibration fits. Nevertheless, the calibration factors derived from the two different pressure campaigns (I) and (II) differed by less than 3.5% and agree within their uncertainty.

It also should be noted that the repeat measurements of samples with 50:50 initial fractions, mentioned at the beginning of this Section 5.3, yielded excellent agreement between samples of about 0.6–0.7%. Note that these repeat data points were included in the plots of Figure 7, but, because of their scatter of less than 1%, were hardly distinguishable on the scale of the graphs in the figure.

### 5.4. Uncertainties in the Measurements and Derived Parameters

For the measurements and derived calibration factors, based on the data shown in Figure 7, three main contributors to the uncertainty can be identified, namely (i) the initial molar fractions of the mixing samples, (ii) the measured Raman signals (integrated Q_1_-bands), and (iii) the calibration curve regression. These are collated in Table 4.

Using only the stable isotopologues H_2_ and D_2_, the uncertainty of the initial molar fractions derived from the mixing vessels pressure reading, total inner volume, and gas purities is ~0.4%. For mixtures containing T_2_, the achievable accuracy of the initial molar ratio is slightly less, with about 0.8–1.0%, due to the lower accuracy in the initial T_2_ gas composition. Note that this is based on a conservative accuracy estimate for the initial T_2_ content.

In order to obtain the normalized Raman band signals for each binary and tertiary sample, one must consider the uncertainty of the measured Raman signals, as well as the influence of the exponential regression described in Equation (7). Therefore, first, the inherent shot-noise of the individual peak intensities must be taken into account; and second, the spectral sensitivity correction as outlined in [25] must be applied. It should be noted that the second contribution is the dominant uncertainty of the integrated Raman band signals (see the spectral intensity calibration data in Figure 1). Both are factored into the subsequent (exponential) regression, as illustrated in Figure 5 above, which is used to derive the Raman band data for the binary and tertiary gas samples.

Note that due to the correlation between the simultaneously fitted calibration curves and the small sample size, a bootstrap resampling method is used [59] to estimate the statistical uncertainty in the calibration curve regression, analogous to the data analysis applied in Ref. [16].

Assessing any systematic uncertainty contributions is rather more involved. By and large, the dominant systematic contribution in TRIHYDE stems from the initial gas purities. In particular, this is true for the T_2_ samples, whose initial concentration value—together with those for the impurities DT and HT—can be sizably different from batch to batch. These then propagate as a systematic variation through a full campaign (each is carried out using just a single T_2_-batch). Varying all initial batch concentration values by their average sample characterization uncertainties, and then applying bootstrapping to the equilibration data again, one can use the difference between deviated and non-deviated results to estimate the systematic uncertainty.

This procedure has been applied to obtain the statistical and systematic uncertainties shown for the calibration factors listed in Table 5.

### 5.5. Application Example: Cross-Calibration for the LARA System Used in KATRIN

It was pointed out earlier (Section 5.2) that the calibration factors $R_{\mathrm{XY}}$ are not of immediate physical or measurement importance; rather it is the ratio between calibration factor $R_{\mathrm{VW},\mathrm{XY}}$ (with VW indicating the isotopologue against which the others are referenced) that becomes useful.

One particularly useful application of these ratio values is the transferability of calibration from one LARA monitoring system to another, provided that the LARA systems have been calibrated for spectral intensity response. Combining the raw calibration factors $R_{\mathrm{XY}}$ obtained in our series of measurement campaigns (Section 5.3 and Section 5.4), we were able to generate a full set of relative isotopologue calibration factors, with the exception of HD; these are collated in Table 5.

Note that all entries are calibration factor ratios, referenced against $R_{H_{2}}$. In order to compare the ratios from all campaigns, all factors were scaled to the reference ratio $R_{H_{2},T_{2}}$ obtained in campaign (I), using H_2_-T_2_ mixtures of 100 mbar. Such reference scaling is always useful (recall the arguments presented in Section 5.3); at times referencing is even necessary, since otherwise it would not be possible to compare data from mixtures without H_2_ content—like campaign (III)—with those from any campaign with H_2_-content—like campaign (II).

The relative calibration factors for all hydrogen isotopologues (with the exception of HD)—as listed in Table 5—may be used in high-precision calibration for a LARA system monitoring of the composition of gas mixtures with tritium content, as , in the case of the KATRIN neutrino mass experiment [3,6].

At present, the LARA monitoring for the T_2_ gas circulating through the KATRIN β-electron source is based on theoretically derived calibration factors, which incorporate the necessary transition probabilities $\Phi_{1,\Delta J}\left(J\prime \prime \right)$ (see Equation (1)), calculated from the transition matrix data of LeRoy [23]. However, in general, it is often difficult to ascertain the absolute calculation accuracy of such transition matrix elements. In the KATRIN experiment the reliability, or “trueness”, of the gas composition is of utmost importance for the evaluation of the neutrino mass. Said transition probabilities were cross-checked in Raman depolarization measurements [54], yielding an agreement of the order 3–4%.

In Figure 8, we compare the theoretical calibration factors $R_{H_{2},\mathrm{XY}}$ used in KATRIN with the experimentally derived TRIHYDE values. It is interesting to note that (i) both series of values are in excellent agreement with each other and (ii) the uncertainties for most of the relative calibration factors are substantially lower than those based on theoretically derived quantities (using Equation (4)). This should not be too surprising since one does not have to rely any longer on the published theoretical Raman transition matrix elements (with relatively large uncertainties [18]) but can exploit our highly accurate gas mixtures for cross-calibration.

## 6. Summary and Conclusions

The aim of the TRIHYDE project has been to set up a facility capable (i) of producing accurate gas mixtures containing tritiated molecules and (ii) of precisely characterizing such mixtures, using a range of mostly in-line gas analysis instrumentation but predominantly exploiting molecule-specific Raman spectroscopy. In this context, at present, TRIHYDE at the Tritium Laboratory Karlsruhe (TLK) constitutes a unique facility, because few—if any—other laboratories have the necessary tritium infrastructure required for tritium handling and sample preparation at their disposal.

In this paper, we have described the setup and operation of TRIHYDE and elucidated the mixing procedures and protocols required to generate precise gas mixtures at chemical equilibrium and to analyze/monitor these samples continuously over extended periods of time.

One particular feature of TRIHYDE is that we have exploited β-induced self-equilibration in tritium-containing mixtures (due to the radioactive decay of tritium) to reach chemical equilibrium on acceptably short time scales (in general <72 h). This is in contrast to the predecessor experiment HYDE for non-radioactive gas mixtures, in which catalytic reactions were required to reach chemical equilibrium in reactive mixtures, such as H_2_-HD-D_2_ evolving from initial H_2_-D_2_ mixtures. It should be noted that TRIHYDE also incorporates a catalytic path for gas circulation; however, for the studies reported here this was not required.

Starting samples for the mixing process could be prepared routinely with a volumetric reproducibility of <0.25%; this was afforded by a combination of reduced volume uncertainty, a streamlined mixing vessel design, and improved pressure diagnostics (in comparison to HYDE). It is worth noting that we used stable isotopologue mixtures of H_2_-D_2_ to confirm our enhanced precision. In addition, the TRIHYDE-integrated mass-spectrometer and binary gas analyzer were used to verify initial gas purities.

The uncertainties contributing to the measurement results with mixtures containing unstable tritium have been addressed in conjunction with the various procedural steps. It is, therefore, worthwhile to summarize these here in order to assess the precision and accuracy of TRIHYDE. As pointed out in Section 5.4, the generation of an individual mixing sample and the determination of its final equilibrated composition incorporates (i) the actual mixing step and (ii) the Raman measurement of the isotopologue fractions in the mixture. Performance data for the TRIHYDE mixing and Raman measurement steps are collated in Table 1, Table 2, and Table 4, and are briefly discussed in the supporting text.

Taking into account only the uncertainties associated with the two “physical” processes of mixing and measuring, one finds, for the former, an uncertainty of 0.50% (for a combined pressure of 500 mbar for the two mixing vessels and excluding the uncertainty related to gas purity) and 0.32% for the determination of individual, integral Raman Q_1_-branch signals. Combined, this yields a potential precision for an individual gas mixture of ~0.60%. In addition, one has to take into account the uncertainty of the T_2_ batch composition, which is dominated by the uncertainty in determining the relative concentration of DT. With this, one expects for the two campaigns’ (T_2_-H_2_ and T_2_-D_2_) values to fall in the range of 0.62–0.83%; indeed, our repeat measurements for 50:50 mixture samples yielded ~0.6–0.8% reproducibility, in line with expectations.

It is worth noting that slight improvement in the precision of the Raman data is possible. At present, it is limited by the accuracy of the Raman intensity calibration with the NIST Raman standard SRM2242, which suffers from a substantial uncertainty at the wavelength of the Q_1_-branch of H_2_ (~1.7%). A work-around for this deficiency would be to add intensity calibration with a second (longer-wavelength) NIST Raman standard SRM2245. Merging the calibrations from these two standards would potentially lower the uncertainties for HD and H_2_ well below the 1% level. However, for excitation, laser radiation at 633 nm is required for the SRM2245 standard. At present, we have explored whether such a dual calibration procedure can be applied without introducing other sources of uncertainty.

As a useful application of TRIHYDE’s accurate mixing and Raman measurement capabilities, we generated a set of Raman analysis calibration factors for all hydrogen isotopologues, which will allow one to verify TRIHYDE mixing samples in other Raman systems and to analyze unknown samples with tritium content with increased confidence. These may serve for cross-validation of calibrated measurement (Raman) instrumentation versus calibrated reference samples. In this context, it is noteworthy that we experimentally confirmed the calibration factors that are used in the gas monitoring of the KATRIN experiment at present (see Section 5.5 above); thus far, for the isotopologues T_2_, DT, and HT these had been based on theoretically calculated factors.

Note that the uncertainties for these derived global calibration quantities (0.8–2.7%—see Table 5 and Figure 8) are larger than the uncertainties for the composition for an individual gas mixture (0.6–0.8%—as noted earlier in this section). This is mainly due to the simultaneous fit to all individual mixture data, as shown in Figure 7, which is needed to derive the calibration factor ratios $R_{H_{2},\mathrm{XY}}$.

Besides the initial task of producing accurate mixing samples with tritium content, and characterizing them, the exploitation of the β-induced self-equilibration process can be further utilized to explore reaction chains and the associated pressure-/temperature-dependent rates. In part, these were explored and validated during this work; however, a full description of the physical and chemical details is well beyond the scope of this paper. For example, the observed equilibrium concentrations agreed rather well with the calculated values, although the concentration dependence of reaction speed showed some unexpected behavior; however, this did not affect final sample quality. These and other topics will be discussed in a forthcoming scientific publication.

Beyond the use of TRIHYDE in producing and characterizing well-defined mixtures of tritiated hydrogen isotopologues from initial dual gas samples, one can envisage a wealth of further applications by expanding the mixing in TRIHYDE to other, non-hydrogen gas components.

For example, we have just commenced experiments in which high-purity CT_4_ is mixed (and equilibrated) with precise molar fractions of H_2_ and D_2_. This should shed light onto the largely unknown spectroscopy of tritium-substituted methanes and provide insight into reaction paths and production rates of these species. These aspects are thought to be of great importance, for example, in tritium–steel surface interactions.

The gas samples that can be prepared in TRIHYDE are not limited to in situ applications. For instance, mixtures of various hydrogen isotopologues were processed using a copper oxide reduction process to generate tritiated water vapor samples. These were then used in high-resolution FTIR studies of HTO [60,61].

Furthermore, precise reference samples, which may include specific “contaminants” (e.g., N_2_) in known quantities might be used to benchmark different methods and perform a cross-calibration of tritium accountancy methods, such as gas chromatography (GC) or beta-induced X-ray spectroscopy (BIXS). Note that the latter technique forms part of the actual TRIHYDE installation.

Finally, one may also consider mixtures with atomic rare gases (not Raman active), such as T_2_ in He, which could be of interest to serve as references in optimizing breeding blanket design for fusion research [62], or T_2_ in Ar, similar to the purge gas in the cryogenic pumping section (CPS) of the KATRIN setup [4].

## Figures and Tables

**Figure 1 sensors-21-06170-f001:**
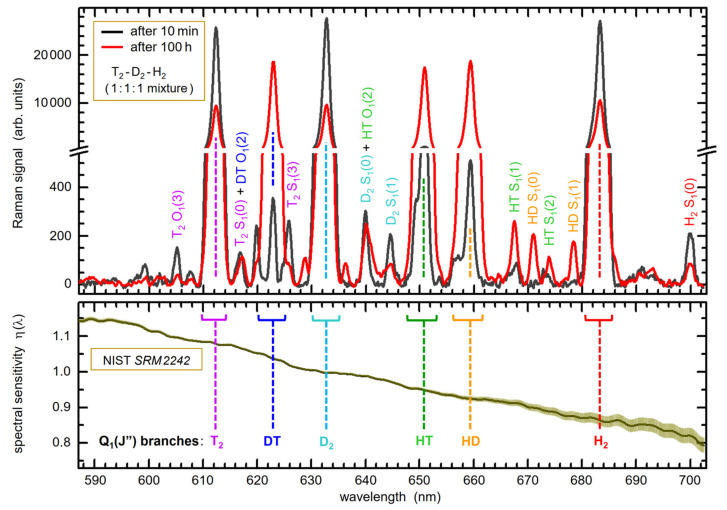
Example Raman spectra of a H_2_-D_2_-T_2_ mixture (ratio 1:1:1) circulating in TRIHYDE, recorded at the beginning (black data trace) and after 100 h of circulation (red data trace); note that the Raman signal data shown here represent averages over 10 Raman measurement cycles of 1 min each. The Q_1_-branches of the six isotopologues, used in the evaluation of concentrations, are annotated (the peak positions by the dashed lines and the band integration intervals by associated brackets). The other features are S_1_- and O_1_-branch lines (to maintain clarity, only a few selected lines are annotated). In the lower panel, the spectral sensitivity calibration for LARA TRIHYDE is shown (normalization at λ = 633 nm); the “raw” Raman spectra were corrected using this response function, resulting in the displayed spectra. For full details, see Section 4.3.

**Figure 2 sensors-21-06170-f002:**
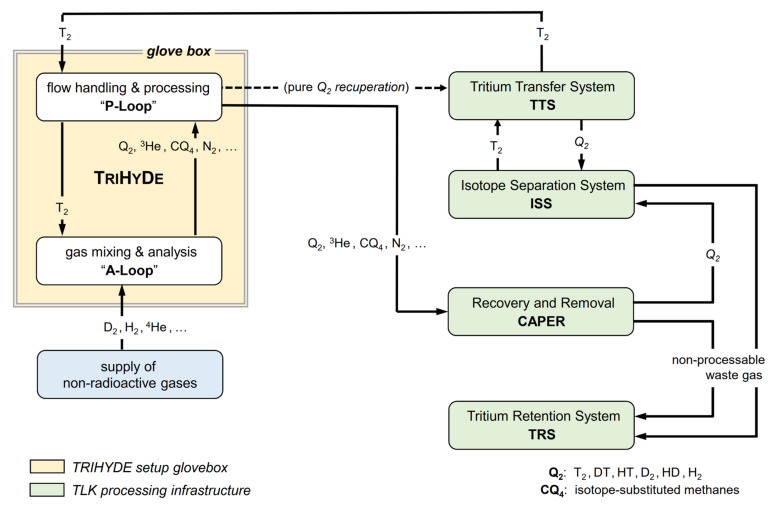
Schematic overview of TRIHYDE and its integration into the TLK infrastructure. TRIHYDE consists of two linked loops for gas mixing and analysis (the A-Loop) and for flow handling/processing of gas mixtures (the P-Loop). Note that CQ_4_ stands for all isotope-substituted methanes, with Q = H, D, T. For details, see text.

**Figure 3 sensors-21-06170-f003:**
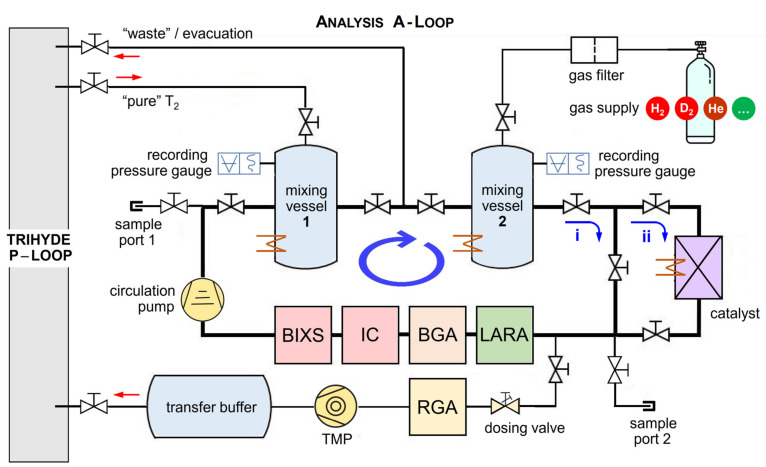
Schematic of the analysis segment of TRIHYDE—the A-Loop. It comprises two volume-calibrated buffer vessels and a series of in-line measurement instruments; optionally, the circulating gas can be directed to pass through a catalyst (to speed up equilibrating/exchange reactions). For compositional analysis: LARA = laser Raman spectroscopy cell, BGA = binary gas analyzer; for activity monitoring: IC = ionization chamber, BIXS = beta-induced X-ray spectrometry unit. In addition, off-line compositional rest gas analysis (RGA) via mass spectrometry is incorporated. Gases can be admitted from the tritium transfer system, from external gas supplied, or via two sample ports. For details, see text.

**Figure 4 sensors-21-06170-f004:**
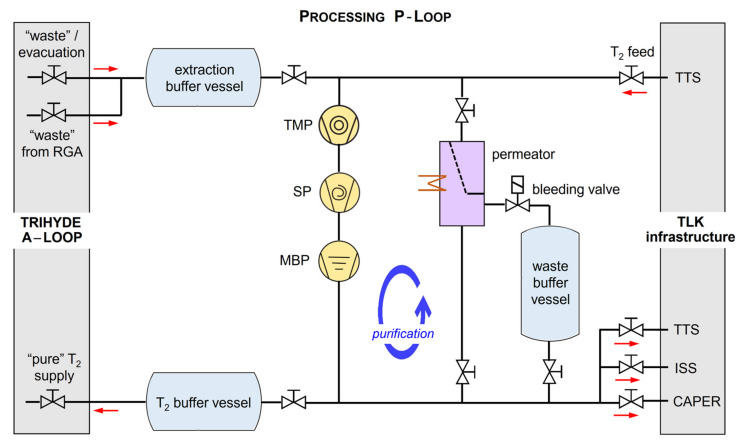
Schematic of the processing segment of TRIHYDE—the P-Loop. Gases enter the loop either from the TTS tritium gas supply or are “waste” gases from the A-loop and exit the loop either to the A-loop or the TLK infrastructure units. The gas (mixtures) are moved and/or compressed by a series of pumps; TMP = turbomolecular pump; SP = scroll pump; MBP = metal bellows pump. Optional to simple gas distribution, the P-loop incorporates the capability of gas cleaning by circulation through a permeator. For details, see text.

**Figure 5 sensors-21-06170-f005:**
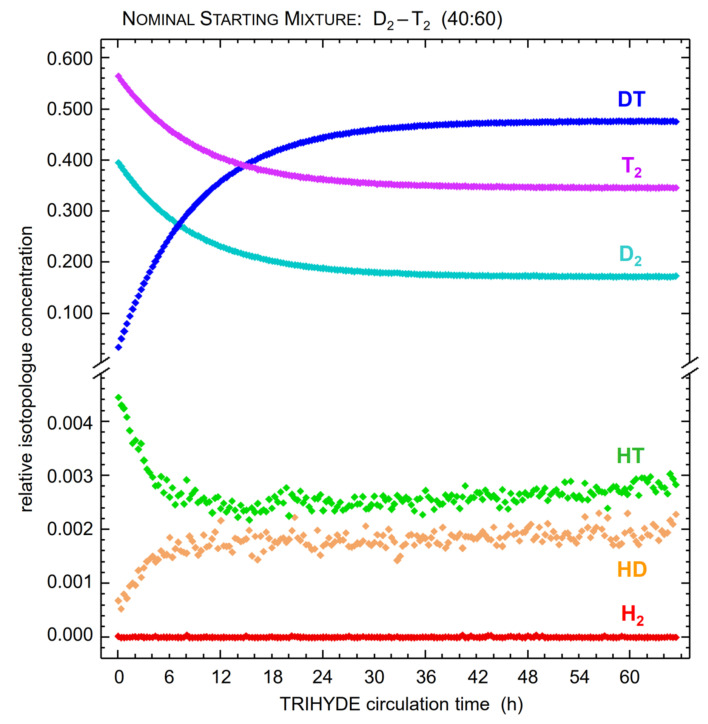
Evolution of the composition of a D_2_-T_2_ sample (nominal starting mixture 40:60), associated with β-induced self-equilibration processes. Note that the T_2_ batch purity in the data shown here was only 97.5%, with most of the remainder being DT. The isotopologue concentrations shown here were extracted from the Raman spectra (signal intensity integrals of the Q_1_-branches); data points represent averages over time intervals of 20 min (corresponding to 20 individual LARA acquisitions). For further details, see text.

**Figure 6 sensors-21-06170-f006:**
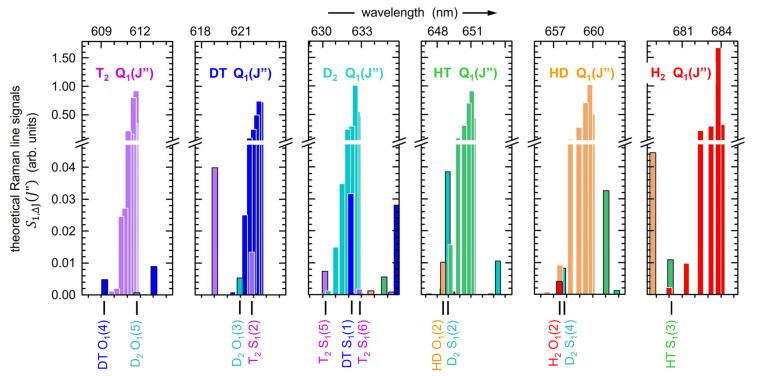
Theoretical Raman signals $S_{1,\Delta J}\left(J\prime \prime \right)\propto \Phi_{1,\Delta J}\left(J\prime \prime \right)\cdot N\left(J\prime \prime ,T\right)$ for all six hydrogen isotopologues, with the transition probabilities $\Phi_{1,\Delta J}\left(J\prime \prime \right)$ based on data provided in Ref. [26]. At the bottom, individual S_1_(*J*″) and O_1_(*J*″) lines are indicated, which might require deconvolution efforts during quantitative evaluation of the Q_1_(*J*″)-branch intensity integrals.

**Figure 7 sensors-21-06170-f007:**
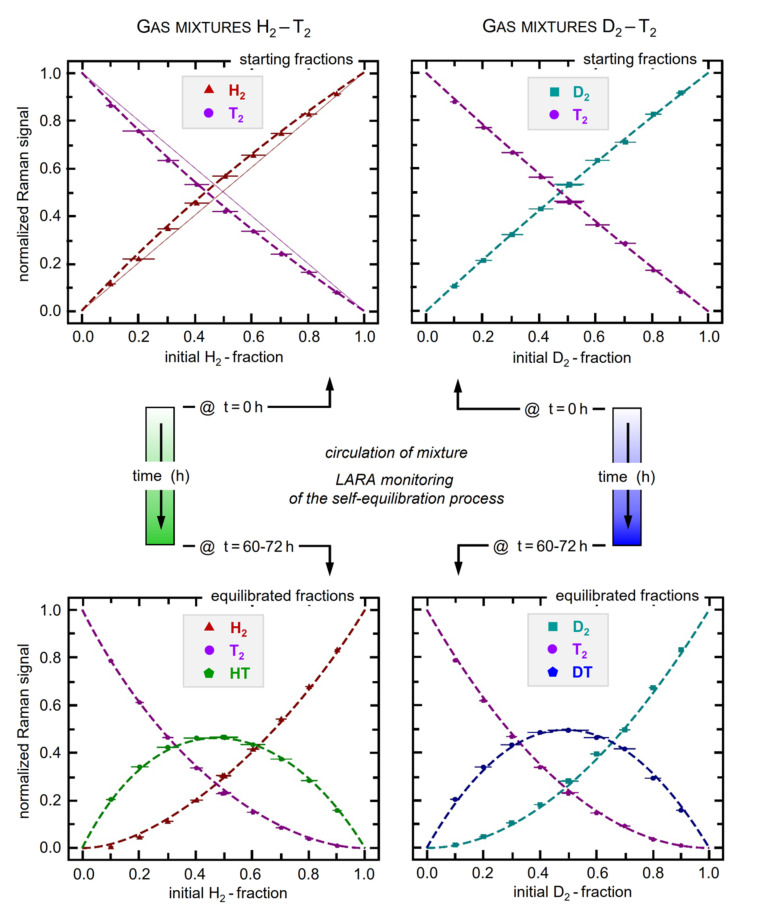
Calibration curves for gas mixtures of H_2_-T_2_ and D_2_-T_2_ (spectral sensitivity corrections and transition probability factors are incorporated in the relative Raman signals). All error bars (1σ) are expanded by a factor of ×5 in the display, for better visibility. Note that the top and bottom panels relate to the data of the initial mixture at t = 0 h and after full equilibration at t > 60 h, respectively. As an example, the straight-line curves for a hypothetical, theoretical intensity relation of $R_{\mathrm{XY}}$ = 1 were added for the initial H_2_-T_2_ mixtures (top left panel). For further details, see text.

**Figure 8 sensors-21-06170-f008:**
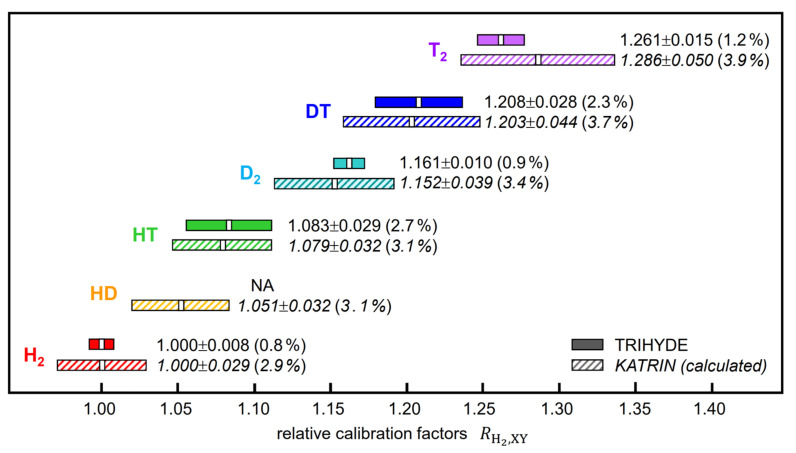
Calibration factor ratios, $R_{H_{2},\mathrm{XY}}$, with X, Y ∈ [H, D, T]; comparison between the experimental values derived in this TRIHYDE work and the calculated (theoretical) values used in the KATRIN experiment, based on the data given in Refs. [16,23].

**Table 1 sensors-21-06170-t001:** Key operating parameters and performance of TRIHYDE and HYDE [16,27].

	TRIHYDE Apparatus		HYDE Apparatus	
Parameter	Capability	Uncertainty	Capability	Uncertainty
Molecular Species	H_2_, D_2_, T_2_, He ^(1)^		H_2_, D_2_	
Pressure Range	1 × 10^−3^–900 mbar	<0.15%	5 × 10^−2^–1000 mbar	0.25%
Temperature Range	20–200 °C	<0.2%	20–300 °C	NA
Mixing Vessel Volume	max. 1600 cm^3^	<0.25%	max. 3320 cm^3^	0.56%

^(1)^ Gases used in this study; range can be extended to other species.

**Table 2 sensors-21-06170-t002:** Summary of key operating parameters and performance data for the LARA monitoring instrumentation.

Parameter	Value	Uncertainty
Operating Laser Power, *I*_L_	1.5–2.0 W	<5 mW
Laser RMS Noise, *I*_noise_		<2 mW
Spectral Range, λ ^(1)^	540–720 nm	
Spectral Calibration, λ		0.07 nm
Spectral Resolution, Δλ ($∆\tilde{\nu }$)	1.05 nm (≅27 cm^−1^)	0.08 nm
Molecular Species	all ^(2)^	
Pressure Range, *p*	1 × 10^−3^–900 mbar	Gauge Reading: 0.12%
Partial Pressure Detection Limit, δ*p* ^(3)^	0.18 mbar	0.08 mbar

^(1)^ Wavelength range governed by spectrometer dispersion and CCD detector size; lower limit dictated by the transmission edge of the Raman edge filter. ^(2)^ In principle, all Raman-active molecular vibrational bands are detectable; their detection limit depends on partial pressure and on Raman transition probability. ^(3)^ For 60 s data accumulation, at 250 mbar loop pressure.

**Table 3 sensors-21-06170-t003:** Purities of the primary gases H_2_, D_2_, and T_2_ used in the various mixing campaigns. Note that the T_2_ batch composition, provided by the ISS of TLK, differed from campaign to campaign, but was nearly stable (change < 0.1%) for the duration of any individual campaign.

	T_2_	DT	D_2_	HT	HD	H_2_
H_2_ (*Linde* Hydrogen N6.0)	---	---	---	---	---	>0.9999
D_2_ (*Linde* Deuterium N5.0; D2.75)	---	---	>0.9950	---	<0.0050	<0.0001
T_2_ (for H_2_ Campaign)	0.994	0.003	<0.001	0.002	<0.001	<0.001
T_2_ (for D_2_ Campaign)	0.975	0.022	<0.001	0.002	<0.001	<0.001

**Table 4 sensors-21-06170-t004:** Overview of typical uncertainty contributions during extraction of the calibration factors, as applicable to the data sets shown in Figure 7.

Process	Parameter	Uncertainty Contribution
Initial Isotopologue Components	Mixing Vessel Volume	0.25%·*V*_MV_
	Mixing Vessel Pressure	0.15%·*p*_MV_
	Mixing Vessel and Loop Temperature	~0.75 °C
	Initial Gas Purity	<0.1% ^(1)^/1.0–2.5% ^(2)^
Raman Measurements	Integrated Q_1_ Bands During Equilibration Process	<0.3%
	Regression Of Equilibration Process (Equation (7))	<0.1%
Calibration Curve Regression	Bootstrap Re-Sampling for Isotopologue Compositions	~0.8% ^(3)^/~2.7% ^(4)^

^(1)^ For inactive gases, H_2_ and D_2_; gas purities listed in Table 3. ^(2)^ Estimate for tritiated mixtures based on initial T_2_ batch measurement. ^(3)^ For “binary” samples with only two isotopologues included. ^(4)^ For “tertiary” samples with three isotopologues included, using fixed chemical equilibrium constants.

**Table 5 sensors-21-06170-t005:** **C**alibration factor ratios $R_{H_{2},\mathrm{XY}}$ , obtained from the TRIHYDE H_2_-T_2_ and D_2_-T_2_ mixing campaigns (II) and (III), respectively, with spectral sensitivity corrections applied; X, Y ∈ [H, D, T].

Isotopologue	$$R_{H_{2},XY}$$	σ_total_	σ_sys_	σ_stat_
H_2_	1.000	0.008	0.003	0.007
HD ^(1)^	---	---	---	---
HT ^(2)^	1.083	0.029	0.015	0.024
D_2_ ^(3)^	1.161	0.010	0.005	0.009
DT ^(3)^	1.208	0.028	0.012	0.026
T_2_ ^(2), (3), (4)^	1.261	0.015	0.008	0.012

^(1)^ Not accessible using only H_2_-T_2_ and D_2_-T_2_ starting mixtures. ^(2)^ Derived from campaign (II). ^(3)^ Derived from campaign (III). ^(4)^ Normalized to $R_{H_{2},T_{2}}$ from campaign (I).

## Data Availability

Selected data available on request from the authors.

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
