# Peer review of "Accurate Reference Gas Mixtures Containing Tritiated Molecules: Their Production and Raman-Based Analysis"

_sensors, 2021, doi:10.3390/s21186170_

Round 1

Reviewer 1 Report

Authors describe a production, handling and analysis facility capable of fabricating gas samples of stable and radioactive hydrogen isotopologues such as D2 and T2. The system is called as “TRIHYDE”. They measure the concentration of a ternary mixed gas such as T2, D2, and DT by Raman spectroscopy, taking their chemical equilibrium into consideration, and show the calibration factors. This finding is expected to be important in experiments that use tritium. The method and results of the experiment are described in great detail, and I don't think there are any major changes required for publication.

  • The calibration curve is shown at the bottom of Figure 1. I think it should be explained how this was determined. Also, it would be good to explain the reason why the error is large on the long wavelength side.

  • There are many itemizations in this manuscript. Therefore, there were some parts that I felt difficult to understand. For example, the bulleted symbols on lines 246-252 also appear on lines 258 and 259, which can be confusing to the readers. I think it is better to delete the parts that do not need bullet points or use other symbols.

  • Regarding the results shown in Fig. 7, it was not clear why the composition of the mixed gas of H2-T2 deviated from the ideal one, and why this did not occur in the D2-T2 system.

  • I found a part that seems to be typographical errors on lines 788 and 801.

Author Response

see attached pdf-file

Reviewer 2 Report

1. The author should mark the position and substance or functional group corresponding to each characteristic peak in the Raman spectrum. 

2. The author introduced a variety of analysis systems, like analysis/measurement A-Loop, LARA analysis system, P-Loop and so on. Please explain and compare. 

3. Data accuracy is a big challenge for SERS testing, and authors need to perform repeatability tests to ensure data reliability. 

4. The author claim that “we have described the setup and operation of TRIHYDE, and elucidated the mixing procedures and protocols to generate precise gas mixtures at chemical equilibrium. and to analyze/monitor these samples continuously over extended periods of time”. Please explain its sensitivity.

5. More information should be provided in the introduction about the significance of this topic. 

6.The format of references is not uniform. Please check the format of the cited documents in this article.

Author Response

see attached pdf-file

Reviewer 3 Report

Dear Editor,

The manuscript sensors-1323494 titled “Accurate reference gas mixtures containing tritiated molecules: their production and Raman-based analysis” by Niemes, Telle and coauthors detailed a TRItium- HYrogen- DEuterium (TRIHYDE) facility capable of preparing gas mixtures of calibrated hydrogen isotopologue contents of H2, HD, D2, T2, DT and HT. The manuscript presents a non-destructive in-line calibration of the gas isotopologue contents with an accuracy on the level of 1% using a home-built laser Raman spectroscopic system.

The facility and designed experiments obvious achieve a high level of success in the production of the special gas mixtures for various applications listed in the introduction part of the manuscript. Due to its potential contribution to some important experiments such as fusion power plant demonstrations ITER and DEMO, and neutrino experiment KATRIN, and its calibration methodology, there would be interest in the manuscript for general readers.

Below are a few comments for the technical content of the manuscript:

  1. For the sake of general interest, are there any other applications or impacts of the tritium related detection in fields/industries other than the three mentioned in the introduction? There are a few examples in the conclusion, but it would be helpful to cite some work other than the fusion/nuclear physics field in the intro.
  2. In the intro and conclusion part, please explain what accuracy is needed/expected in these sample preparation as well as what has been achieved. In the abstract, a level of 1% accuracy is mentioned, but not clearly referred in the main text except a 0.8-1.0% in Line 784.
  3. Equation (4) is not straight forward enough. What is its relation to Eqn. (1)? Is R_XY,exp average value of ?(?_Raman) over a small range of wavelength? Does R_XY,theory contains every parameter of the response function in Eqn. (1) except ?(?_Raman)?
  4. The initial gas mixtures have been characterized to a higher accuracy than .1%, is there a room for improvement for the in-line LARA system measurement accuracy and/or other components of the experimental system? This should be discussed in the conclusion.
  5. Some acronyms are not clearly spell out for general readers when first used, e.g., ITER/DEMO, CAPER

Best regards,

Referee

Author Response

see attached pdf-file
